# Preparation and Properties of Atomic-Oxygen Resistant Polyimide Films Based on Multi-Ring Fluoro-Containing Dianhydride and Phosphorus-Containing Diamine

**DOI:** 10.3390/polym16030343

**Published:** 2024-01-26

**Authors:** Zhenzhong Wang, Xi Ren, Yan Zhang, Changxu Yang, Shujun Han, Yuexin Qi, Jingang Liu

**Affiliations:** Engineering Research Center of Ministry of Education for Geological Carbon Storage and Low Carbon Utilization of Resources, School of Materials Science and Technology, China University of Geosciences, Beijing 100083, China; 3003230013@email.cugb.edu.cn (Z.W.); renxi@email.cugb.edu.cn (X.R.); 3003200016@email.cugb.edu.cn (Y.Z.); 2103220040@email.cugb.edu.cn (C.Y.); 2103230036@email.cugb.edu.cn (S.H.); 2003230014@email.cugb.edu.cn (Y.Q.)

**Keywords:** fluoro-containing polyimide, atomic oxygen, low earth orbit, optical properties, thermal properties

## Abstract

Colorless and transparent polyimide (CPI) films with good atomic oxygen (AO) resistance and high thermal endurance are highly required in low earth orbit (LEO) space exploration. Conventional CPI films based on fluoro-containing 4,4′-(hexafluoroisopropylidene)diphthalic anhydride (6FDA) have been widely used in space applications. However, the AO erosion yields and glass transition temperatures (T_g_) of the 6FDA-based CPI films have to be modified in order to meet the severe serving environments. In the current work, novel CPI films based on a multi-ring fluoro-containing 9,9-bis(trifluoromethyl)xanthene-2,3,6,7-tetracarboxylicdianhydride (6FCDA) monomer were developed. In order to enhance the AO resistance of the derived CPI film, a phosphorus-containing aromatic diamine, 2,5-bis[(4-aminophenoxy)phenyl]diphenylphosphine oxide (BADPO) was used to polymerize with the dianhydride to create the organo-soluble resin. Then, two phosphorus-containing CPI films (PPI), including PPI-1 (6FDA-BADPO) and PPI-2 (6FCDA-BADPO) were prepared by thermally curing of the PPI solutions at elevated temperatures. The PPI films maintained good optical transparency with transmittance values over 80% at a wavelength of 450 nm. PPI-2 exhibited a T_g_ value of 311.0 °C by differential scanning calorimetry (DSC) measurement, which was 46.7 °C higher than that of the PPI-1 counterpart (T_g_ = 264.3 °C). In addition, the PPI-2 film showed a coefficient of linear thermal expansion (CTE) value of 41.7 × 10^−6^/K in the range of 50~250 °C, which was apparently lower than that of the PPI-1 sample (CTE = 49.2 × 10^−6^/K). Lastly, both of the two PPI films exhibited good AO resistance with the erosion yields (E_y_) of 6.99 × 10^−25^ cm^3^/atom for PPI-1 and 7.23 × 10^−25^ cm^3^/atom for PPI-2 at an exposure flux of 5.0 × 10^20^ atoms/cm^2^. The E_y_ values of the current PPI films were obviously lower than that of the standard polyimide (PI) film based on pyromellitic dianhydride (PMDA) and 4,4′-oxydianiline (ODA) (E_y_ = 3.0 × 10^−24^ cm^3^/atom).

## 1. Introduction

Aerospace exploration has been gaining great progress in recent decades thanks to the application of various new materials [1,2,3]. Continuous research and development (R&D) of new materials plays an important role in improving the reliability of spacecraft in severe space environments [4]. Polymers and polymer-based composites have been paid increasing attention in the R&D of new space materials due to their low density, light weight, high strength, and excellent processability [5]. Polyimide (PI) represents a class of high-performance polymers characterized by the highly conjugated and polar five- or six-membered heteroaromatic imide rings in its molecular structures [6,7,8,9,10]. The structural features endow PIs with excellent combined properties, including extremely wide servicing temperatures from −269 °C to 400 °C, excellent tensile and dielectric properties, excellent radiation (ultraviolet, X-ray, neutron, proton, electron, gamma-ray, etc.) resistance, and so on [11]. More importantly, the properties of PIs can be modified so as to meet the different space applications due to the ability to flexibly design the molecular structures of the PIs. Thus, various PIs have been successfully applied in the space exploration for more than half a century, since the first commercialization by DuPont Company, Wilmington, DE, USA in the mid-1960s [12]. Standard PIs, such as Kapton^®^ films, Vespel^®^ moldings, and Pyre-ML^®^ enamels (all registered trademarks of DuPont, USA) are usually based on pyromellitic dianhydride (PMDA) and 4,4′-oxydianiline (ODA). This repeating unit is known for its high conjugation, high polarity, and high charge transfer (CT) interactions from the ODA part as electron-donator to the PMDA part as electron-acceptor [13,14,15]. This structural feature endows PI with excellent heat resistance and mechanical properties, but also inevitably causes some property defects in standard PIs, such as darker color. Meanwhile, the molecular chains of conventional PIs are susceptible to degradation by atomic oxygen (AO) in low earth orbit (LEO), vacuum ultraviolet (VUV), and charged particle irradiation in geosynchronous earth orbit (GEO) and deep space [16,17,18,19,20]. This undoubtedly hinders the applications of standard PIs in the field of space exploration. Thus, functional PI films have been widely studied and developed so as to meet the property requirements for LEO, GEO, or deep-space applications. For LEO applications, such as the multi-layer thermally insulating (MLI) blankets for satellites, thermal shielding layers for space telescopes, substrates for solar cells, and so on, the PI films should have excellent AO resistance besides their excellent thermal and mechanical properties. While for GEO or deep-space applications, such as the membranes for solar sails, the optical components for satellites, and so on, PI films should have excellent optical transparency, high thermal and dimensional stability, and good long-term reliability [12].

In the 1980s, two fluoro-containing PI (FPI) films were developed at the Langley Research Center (LaRC), National Aeronautics and Space Administration (NASA), USA, and named as LaRC-CP1 and LaRC-CP2, respectively [21,22]. The two FPI films are based on fluoro-containing 6FDA dianhydride and the former is derived from the fluoro-containing diamine 2,2-bis [4-(4-aminophenoxy)phenyl]hexafluoropropane (BDAF) and the latter from 1,3-bis(3-aminophenoxy)benzene (133APB), respectively [23]. The two FPI films exhibited excellent optical transparency, ultraviolet resistance, good moisture resistance, low dielectric constants, and good solution-processability. The excellent combined properties of the two FPI films made them good candidates as substrates for solar cells, membranes for large inflatable antenna, solar sails, and other deployable structures for space applications. LaRC-CP1 film has been proven to be stable in GEO space environments for more than 10 years [24]. However, these two FPI films are AO sensitive and are easily eroded in LEO space environments. In addition, the T_g_ values of the two FPI films (T_g_ = 264 °C for LaRC-CP1 and T_g_ = 207 °C for LaRC-CP2) are not high enough to meet some severe space applications [24]. Thus, the FPI films have to be modified to enhance their AO resistance and high-temperature durability.

In the 1990s, scientists at DuPont first reported FPIs based on a novel fluoro-containing dianhydride, 9,9-bis(trifluoromethyl)xanthene-2,3,6,7-tetracarboxylic dianhydride (6FCDA) [25,26,27,28]. Compared with the FPIs derived from un-cyclized 6FDA, the multi-ring structure in 6FCDA endowed the FPIs with various specific features, such as reduced solubility in organic solvents, improved solvent resistance, reduced water uptakes, enhanced T_g_, reduced coefficients of linear thermal expansion (CTE), and so on. Most of the properties are required for high-tech applications, such as microelectronic packaging, electrical insulating, and also aerospace exploration. Various common aromatic diamines, including 4,4′-oxydianiline (ODA), p-phenylenediamine (PPD), diaminodurene (DAD), 3,3′,5,5′-tetramethylbenzidine (TMB), 2,2′-dimethylbenzidine (DMB), 2,2′-bis(trifluoromethyl)benzidine (TFMB), and bis(4-aminophenyl)hexafluoropropane (6FDAm) were used for the development of 6FCDA-PIs. Nevertheless, to the best of our knowledge, until today, the application of 6FCDA dianhydride in FPIs has not been more extensively studied, and the structure-property relationship of the 6FCDA-based FPIs with functional diamines remains to be revealed more deeply and systematically.

Thus, in the current work, a new PI film based on 6FCDA and a phosphorus-containing diamine, 2,5-bis[(4-aminophenoxy)phenyl]diphenylphosphine oxide (BADPO) was first prepared, which was named as PPI-2. For comparison, the referenced PPI-1 was prepared from 6FDA and BADPO according to established procedure. The purpose of the current work is to develop functional FPI films with potential applications in LEO space environments. The phosphorus-containing diamine might endow the FPI films with enhanced atomic-oxygen resistance, while the 6FCDA dianhydride might improve the glass transition temperatures (T_g_) and decrease the linear coefficients of thermal expansion (CTE) of the FPI films. Effects of the dianhydride and diamine structures on physical and chemical properties, especially the thermal, optical, and AO erosion behaviors of the PI films, were investigated in detail.

## 2. Materials and Methods

### 2.1. Materials

9,9-Bis(trifluoromethyl)xanthene-2,3,6,7-tetracarboxylic dianhydride (6FCDA, purity: 99.5%) and 4,4′-(hexafluoroisopropylidene)diphthalic anhydride (6FDA, purity: 99.6%) were commercially available from ChinaTech Chem. Co. Ltd. (Tianjin, China) and dried in vacuo at 180 °C for 24 h before use. The diamine of 2,5-bis[(4-aminophenoxy)phenyl]diphenylphosphine oxide (BADPO, purity: 99.5%) was synthesized and purified in our laboratory according to procedures established in [29]. The ultra-dry (purity: 99.9%, water content < 50 ppm) organic solvents of N,N-dimethylformamide (DMF), N,N-dimethylacetamide (DMAc), N-methyl-2-pyrrolidone (NMP), acetic anhydride (Ac_2_O), and pyridine were bought from Innochem Science & Technology Co., Ltd. (Beijing, China) and used as received. The other analytical reagents, including deuterated dimethyl sulfoxide (DMSO-d_6_) and deuterated DMF-d_7_, were obtained from Innochem Science & Technology Co., Ltd. (Beijing, China) and used directly. The referenced Kapton^®^ type of film, poly(pyromellitic dianhydride-co-4,4′-oxydianiline) (PMDA–ODA) was gift from RAYITEK Hi-Tech Film Company, Co., Ltd. (Shenzhen, China) and used for AO tests.

### 2.2. Characterization Methods

For PPI resins, the inherent viscosities were tested with an Ubbelohde viscometer (Shanghai Bilon Instrument Co., Ltd., Shanghai, China) with a 0.5 g/dL NMP solution at room temperature. The number average molecular mass (M_n_) and weight average molecular mass (M_w_) were detected with a gel permeation chromatography (GPC) system (Shimadzu, Kyoto, Japan) with the HPLC grade of NMP as the mobile phase. The polydispersity index (PDI) of the molecular mass was calculated as PDI = M_w_/M_n_. Hydrogen nuclear magnetic resonance (^1^H-NMR) was tested on an AV 400 spectrometer (Bruker, Ettlingen, Germany) operating at 400 MHz in DMSO-d_6_ for PPI-1 and DMF-d_7_ for PPI-2. Wide-angle X-ray diffraction (XRD) of the PPI powders was conducted on a M03X-HF X-ray diffractometer (Bruker AXS, Karlsruhe, Germany) with Cu-Kα1 radiation operated at 40 kV and 200 mA. Solubility was investigated by mixing 1.0 g of the PPI resin and 9.0 g of the tested solvent to afford a mixture with 10 wt% solid content. Then, the mixture was stirred for 24 h at room temperature. The solubility was determined visually as three grades: completely soluble (++), partially soluble (+), and insoluble (−).

For PPI films, the Fourier transform infrared (FTIR) spectra were recorded on an Iraffinity-1S FT-IR spectrometer (Shimadzu, Kyoto, Japan). Ultraviolet-visible (UV-Vis) spectra of the PPI films were collected on a U-3210 spectrophotometer (Hitachi, Tokyo, Japan). The in-plane refractive indices (n_TE_) and out-of-plane refractive indices (n_TM_) were measured with a Model 2010/M prism coupler (Metricon Company, Pennington, NJ, USA) at the wavelength of 632.8 nm. The average refractive indices (n_av_) were calculated as n_av_ = [(2n_TE_^2^ + n_TM_^2^)/3]^1/2^. The birefringence (Δn) were calculated as Δn = n_TE_ − n_TM_. CIE (International Commission on Illumination) optical parameters, including L* (lightness, 100 means white and 0 indicates black), a* (positive one means a red color, and a negative one indicates a green color) and b* (positive one means a yellow color, and a negative one indicates a blue color) were tested with a color i7 spectrophotometer (X-rite Company, Grand Rapids, MI, USA) at a thickness of 50 μm. Thermogravimetric analysis (TGA) and the derivative TGA (DTG) were performed on a TG 209F3 thermal analysis system (NETZSCH, Selb, Germany) at a heating rate of 20 °C/min in nitrogen. Differential scanning calorimetry (DSC) was carried on a DSC 200 F3 Maia system (NETZSCH, Selb, Germany) at a heating rate of 10 °C/min in nitrogen. Dynamic mechanical analysis (DMA) was recorded on a TA-Q800 thermal analysis system (New Castle, DE, USA) at a heating rate of 5 °C/min and a frequency of 1Hz in nitrogen. Thermo-mechanical analysis (TMA) was recorded on a TMA402F3 thermal analysis system (NETZSCH, Selb, Germany) with the temperatures from 50~450 °C at a heating rate of 5 °C/min in nitrogen atmosphere. The coefficients of linear thermal expansion (CTE) values of composite films were recorded in the range of 50~250 °C. Tensile properties, including tensile strength (T_S_), tensile modulus (T_M_), and elongations at break were detected with an Instron 3365 universal test machine (Instron Corp., Norwood, MA, USA) with the sample size of 80 mm (length) × 10 mm (width) × 0.05 mm (thickness) at a drawing rate of 2.0 mm/min. The dielectric properties, including dielectric constant (D_k_) and dielectric dissipation factor (D_f_) at the frequency of 10 GHz, were measured by Agilent E5063A vector network analyzer (Agilent Technologies Company, Palo Alto, CA, USA) at room temperature.

The atomic oxygen (AO) erosion performance of PPI films was obtained in a ground-based AO effects simulation facility in Beijing Institute of Spacecraft Environment Engineering [30]. The size of PPI film sample was 20 (length) × 20 (width) × 0.05 (thickness) mm^3^. The AO fluence for the test was 5.0 × 10^20^ atoms/cm^2^, and the mass loss of PPI films after AO exposure was recorded. The erosion yield (E_y_) of the PPI samples is calculated by the following Equation (1) [30]:(1)$$E_{y}=\frac{∆M_{s}}{A_{s}\rho_{s}F}$$
where, E_y_ = erosion yield of the sample (cm^3^/atom); Δ$M_{s}$ = mass loss of PPI sample (g); A_s_ = surface area of the PPI sample exposed to AO (cm^2^); ρ_s_ = density of the sample (g/cm^3^); F = AO fluence (atoms/cm^2^). As standard Kapton^®^ film has a specific erosion yield of 3.0 × 10^−24^ cm^3^/atom [30], and all the current PPI samples in the test are supposed to possess similar density and exposure area with the referenced Kapton^®^ film, therefore, the simplified Equation (2) could be used to calculate the E_y_ of the PPI films:(2)$$E_{y}=\frac{∆M_{s}}{∆M_{Kapton}}E_{Kapton}$$
where, E_Kapton_ was the E_y_ value of the Kapton^®^ film (3.0 × 10^−24^ cm^3^/atom); ΔM_Kapton_ was the mass loss of Kapton^®^. The surface elemental composition of the PPI films were tested by X-ray photoelectron spectroscopy (XPS) with an ESCALab220i-XL electron spectrometer (Thermo Fisher Scientific, Waltham, MA, USA), under the monochromatic MgKα radiation. The surface morphology was obtained on a Technex Lab Tiny-SEM 1540 field-emission scanning electron microscope (FE-SEM) (Tokyo, Japan) with an accelerating voltage of 15 KV for imaging. Atomic force microscopy (AFM) was operated on a Multimode 8 AFM microscope (Bruker, Santa Babara, CA, USA) with tapping mode. The Scanasyst-Air model of cantilever (Bruker, Santa Babara, CA, USA) with the resonance frequency (f_0_) of 70 kHz, spring constant (k) of 0.4 N/m and the size of 115 μm × 25 μm × 650 nm) was used.

### 2.3. PPI Resins Synthesis and Films Preparation

The PPI resins were prepared from BADPO and 6FDA or 6FCDA via a two-step chemical imidization procedure, which could be illustrated by the synthesis of PPI-2 (6FCDA-BADPO). Into a three-necked 500 mL flask equipped with a mechanical stirrer, a nitrogen inlet, and a cold water bath was added BADPO (24.6250 g, 50 mmol) and dry NMP (120.0 g). The clear BADPO solution was obtained after stirring at 10~15 °C for 10 min under nitrogen. Then, the 6FCDA white powder (22.9110 g, 50 mmol) was added into the flask with one batch and additional NMP (22.6 g) was added into the reaction system to wash the residual 6FCDA in the adding funnel. By this way, all the solid monomers were transferred into the flask and a solid content of 25 wt% was achieved at the same time. The cold bath was removed after 1 h and the reaction mixture remained as suspension even after stirring at room temperature for 3 h, indicating the low solubility of the 6FCDA in the reaction medium. After another 20 h, the viscous and homogeneous pale-yellow poly(amic acid) (PAA) solution was obtained. Then, the dehydrating agent of acetic anhydride (30.6 g, 300 mmol) and catalyst of pyridine (19.8 g, 250 mmol) were added into the PAA solution with vigorous stirring. The chemical imidization procedure was carried out for 24 h at room temperature. Then, the reaction mixture was slowly added into aqueous ethanol solution (2 L, 75 vol%). The PPI-2 resin was obtained as filaments in the precipitating medium. The resin was thoroughly immersed into the ethanol solution and then collected by filtration. The wet resin was first dried in a fume hood overnight at room temperature and then further dried in vacuum at 120 °C for 10 h to remove the liquids. At last, dry and pale-yellow PPI-2 resin was obtained. Yield: 44.6 g (97.6%). M_n_: 9.61 × 10^4^ g/mol; M_w_: 15.27 × 10^4^ g/mol; PDI: 1.59. ^1^H-NMR (DMF-d_7_, ppm): 8.58 (s, 2H), 8.26–8.21 (m, 2H), 8.07–8.03 (m, 4H), 7.86–7.69 (m, 10H), 7.52–7.50 (m, 4H), 7.38–7.36 (m, 1H), and 7.19–7.17 (m, 2H).

PPI-1(6FDA-BADPO) resin was prepared according to the procedure mentioned above with the formula shown in Table 1. M_n_: 26.74 × 10^4^ g/mol; M_w_: 33.74 × 10^4^ g/mol; PDI: 1.26. ^1^H-NMR (DMSO-d_6_, ppm): 8.19–8.15 (m, 2H), 7.95 (s, 2H), 7.74–7.68 (m, 6H), 7.56–7.38 (m, 10H), 7.32–7.30 (m, 4H), 7.23–7.20 (d, 2H), 7.07–7.03 (m, 1H), and 6.83–6.81 (d, 2H).

The fully dried PPI-2 resin was dissolved in ultra-dry DMAc at a solid content of 15 wt%. The obtained crude PPI-2 solution was purified by filtration through a 1.0 μm Teflon syringe filter. The impurities were removed and the afforded PPI-2 solution was de-foamed in a desiccator at room temperature for 10 h. The homogeneous PPI-2 solution was cast onto a clean borosilicate glass (150 mm × 100 mm × 3 mm) with an automatic film applicator (AFA-II, Shanghai Meiyu Equipment Co., Ltd., Shanghai, China) and the thickness of the pristine liquid films were controlled by adjusting the gap distances of the casting bar. Then, the glass substrates were thermally dried in a programmable oven with nitrogen environments by the heating procedure of 80 °C/3 h, 150 °C/1 h, 180 °C/1 h, and 250 °C/1 h, respectively. After the thermal treatments, the cooled glass substrates were then immersed into the deionized water. The PPI-2 film peeled off the substrate and was obtained as a free-standing film with pale-yellow color. After drying at 120 °C for 3 h, the PPI-2 film was evaluated for various properties.

PPI-1 film was prepared according to the similar procedure except PPI-2 resin was replaced by PPI-1 resin.

## 3. Results and Discussion

### 3.1. PPI Resins Synthesis and Films Preparation

As shown in Figure 1, two phosphorus-containing PI (PPI) resins were prepared via a two-stage chemical imidization procedure. Homogeneous PPI solutions were obtained after in situ imidization in the same reactor, reflecting the good solubility of the afforded PPI resins. Flexible and tough PPI resins were obtained as continuous filaments, indicating the high molecular mass of the polymers. The inherent viscosities ([η]_inh_), molecular mass and the solubility of the PPI resins were detected and are summarized in Table 2. Basically, the PPI-2 resin showed inferior [η]_inh_ and molecular mass values than those of the PPI-1 counterpart. For example, PPI-2 resin showed [η]_inh_ and M_w_ values of 1.21 dL/g and 15.27 × 10^4^ g/mol, respectively, which were lower than those of the PPI-1 ([η]_inh_ = 1.38 dL/g; M_n_ = 33.74 × 10^4^ g/mol) (Figure 2). As we know, the molecular mass of one PI is highly related with the purities and reactivities of the polymerization monomers, and also affected by the microscopic stacking states of the molecular chains. Considering the same diamine of BADPO was used for both of the PPI systems, these physical parameters were mainly determined by the feature of the dianhydrides. The two fluoro-containing dianhydrides possessed the same level of purities (99.5~99.6%) according to the technical data sheet (TDS) provided by the supplier.

As for the polymerization reactivity, it has been well established in the literature that the lowest unoccupied molecular orbital (LUMO) energy levels (ε_LUMO_) of one dianhydride were closely related with their polymerization reactivity due to the nucleophilic reaction nature for the formation of poly(amic acid) [31]. A lower ε_LUMO_ value for the dianhydride usually indicates higher reactivity of the monomer [32]. In the current work, the ε_LUMO_ values of the dianhydrides were calculated according to the density functional theory (DFT)/B3LYP methods with Gaussiansoftware (Version: 09, Gaussian, Wallingford, CT, USA) using the 6-311G(d, p) basis set [33] and the results are shown in Figure 3. According to the simulation results, 6FCDA showed the ε_LUMO_ value of −3.56 eV, which was very close to that of the 6FDA analogue (ε_LUMO_ = −3.42 eV). This indicates that the two dianhydrides showed quite similar reactivity.

Besides the ε_LUMO_ values of the dianhydrides, Figure 3 also labels the calculated torsion angles of the dianhydrides, which was helpful for predicting the molecular chains conformation and packing information. For 6FDA, the calculated bridge-bond angle (θ_1_) between the two anhydride residues (C-C-C) was 111.3°, which was very closed with the computer-modelling data for 6FDA-based model compounds in the literature (112°) [33]. Comparatively, the 6FCDA showed a similar angle (θ_2_) of 111.9° with that of 6FDA. However, for 6FCDA, another bridge-bond angle (θ_3_) between the two anhydride residues (C-O-C) was 121.1°. This indicates that the xanthene unit in the 6FCDA danhydride is nearly planar with its central ring [34]. Thus, the molecular chains in PPI-2 might form more ordered packing than that of PPI-1. This could be deduced from the inter-segmental spacing (d) results calculated by Bragg’s equation of nλ = 2d × sin θ, where n is the order of reflection (n = 1), λ is the wavelength of incident radiation (1.54 Å), and θ is the scattering angle obtained by the XRD measurements (Figure 4a) [35]. PPI-2 showed a d value of 0.521 nm, which was lower than that of its PPI-1 counterpart (d = 0.543 nm). Thus, microscopically, the molecular chains of PPI-1 exhibited highly entangled state, while those of PI-2 exhibited a locally ordered stacked structure. This might be the reason why PPI-1 exhibited higher molecular mass in the GPC tests [36].

The solubility of the two PPI resins in five representative solvents were detected and the results are shown in Table 2. Both of the PPI resins were soluble in polar aprotic solvents of NMP, DMAc, and DMF at room temperature at a solid content of 10 wt%. This is mainly because the flexible hexafluoroisopropylidene units in the dianhydride moiety and the lateral bulky diphenylphosphine oxide (DPPO) substituents in the diamine moiety efficiently decreased the packing density of the molecular chains in the polymers. PPI-1 was also soluble in DMSO and CPA, while PPI-2 was partially soluble in DMSO and not soluble in CPA. The reduced solubility of PPI-2 was mainly ascribed to the relatively rigid molecular skeleton in the 6FCDA units. The enhanced solvent resistance for the 6FCDA-based PIs is quite beneficial for the practical applications, especially in some solvent-sensitive fields.

The chemical structures of the PPI resins were confirmed by the ^1^H-NMR measurements and the results are shown in Figure 4b. Two different deuterated solvents were used for the ^1^H-NMR measurements due to the different solubility of the two samples. According to the ^1^H-NMR spectra of PPI-1 in DMSO-d_6_ and PPI-2 in DMF-d_7_, the protons adjacent to the highly electron-withdrawing imide carbonyl groups in the dianhydride moiety (H_1_ and H_2_) showed the absorptions at the farthest downfield in the spectra. The protons adjacent to the imide carbonyl groups in the diamine moiety (H_a_ and H_a′_) showed the absorptions at the second farthest downfield in the spectra. Contrarily, the protons adjacent to the electron-donating ether groups in the diamine moiety (H_c_ and H_d_) exhibited the absorptions at the farthest upfield in the spectra. The revealed structural information is in good agreement with the theoretical ones and thus confirmed the formation of the targeted PPI resins.

Flexible and tough PPI films were prepared from the corresponding PPI solutions in DMAc at elevated temperatures up to 250 °C. By the high-thermal curing procedure, the DMAc solvent in the PPI films were totally removed. Figure 5 shows the FTIR spectra of the PPI films, together with the assignments of the characteristic absorption peaks. The two samples exhibited similar absorptions due to the similar chemical structures. In the spectra, the characteristic absorptions of the imide rings, including the asymmetrical carbonyl stretching vibrations at 1778 cm^−1^, the symmetrical carbonyl stretching vibrations at 1719 cm^−1^, the carbonyl bending vibrations at 719 cm^−1^, and the C–N stretching vibrations at 1383 cm^−1^ were all observed. In addition, the characteristic absorptions of the C-F bonds at 1102 cm^−1^, the C=C bonds at 1496 cm^−1^, the C-O-C bonds at 1211 cm^−1^, and the P=O bonds at 1161 cm^−1^ were also detected. The reflected structural features are in consistent with the anticipated polymers.

### 3.2. Thermal Properties

The thermal stability of the PPI films were evaluated and the results are shown in Table 3. Figure 6 depicts the TGA and DTG plots of the PPI films. Both PPI films exhibited good thermal stability before 450 °C, after which the thermal decomposition occurred. The 5% weight loss temperatures (T_5%_) were recorded at 537.6 °C for PPI-1 and 501.3 °C for PPI-2, respectively. A two-stage decomposition behavior was observed for both PPI films according to the DTG plots. The first degradation recorded at the maximum decomposition temperature (T_max1_) of 550.5 °C for PPI-1 and 516.0 °C for PPI-2 was attributed to the loss of the lateral DPPO substituents in the polymers. Then, the second decomposition was found at the maximum decomposition temperature (T_max2_) of 611.1 °C for PPI-1 and 600.1 °C for PPI-2, which was due to the degradation of the main chains of the polymers. At 750 °C, PPI-1 and PPI-2 showed the residual weight ratio (R_w750_) values of 59.9% for PPI-1 and 63.4% for PPI-2, respectively. Basically, PPI-1 showed a slightly higher thermal stability than that of its PPI-2 counterpart. This might be due to the existence of the thermally sensitive -C-O-C- bridge in the 6FCDA units.

Figure 7 and Figure 8 show the DSC and DMA plots of the PPI films, respectively. From the plots, the T_g_ values of the PPI films were revealed. PPI-1 and PPI-2 showed the T_g_ values of 264.3 °C and 311.0 °C by DSC measurements, respectively. Similarly, the two PPI films showed the T_g_ values of 274.2 °C for PPI-1 and 316.9 °C for PPI-2 by DMA tests, respectively. Apparently, in both tests, PPI-2 showed obviously enhanced T_g_ values compared to those of the analogous PPI-1 film. The relatively higher molecular chain packing density in PPI-2 efficiently prohibited the free motion of the chain segments, which in turn increased the T_g_ of the polymer. The results indicate that the molecular design of improving the T_g_ values of the fluoro-containing PI films via incorporation of the rigid xanthene units is feasible.

Incorporation of the multi-ring xanthene units in 6FCDA also improved the high-temperature dimensional stabilities of the PPI-2 film, which could be deduced from the TMA measurements shown in Figure 9. The PPI-2 film showed the CTE value of 41.7 × 10^−6^/K in the temperature range of 50 to 250 °C, which was apparently lower than that of the PPI-1 (CTE = 49.2 × 10^−6^/K). The low-CTE feature for the PPI-2 film is quite beneficial for the practical applications.

The tensile properties of the PPI films were detected and the tensile strength (T_S_), tensile modulus (T_M_) and elongations at break (ε) values of the samples are shown in Table 3. As expected, the PPI-2 film based on the rigid 6FCDA dianhydride showed the higher T_M_ value (4.05 GPa) than that of the PPI-1 film derived from the flexible 6FDA dianhydride (T_M_ = 3.54 GPa). In addition, the PPI-2 film showed T_S_ and ε values of 125.5 MPa and 9.9%, respectively, which were also superior to those of its PPI-1 counterpart (T_S_ = 115.1 MPa; ε = 6.4%). The condense molecule chain packing in PPI-2 efficiently increased the intermolecular interactions, which improved the tensile properties of the polymer.

In summary, the currently developed PPI-2 film showed higher T_g_, higher T_S_ and higher T_M_ values than those of the fluoro- and phosphorus-containing PI films reported in the literature, such as the one derived from 6FDA and 2,5-bis[(4-amino-3-trifluoromethylphenoxy)phenyl]diphenylphosphine oxide (T_g_ = 271 °C; T_S_ = 124.9 MPa; T_M_ = 1.9 GPa) [37] or the one from 6FDA and bis(3-aminophenyl)-3,5- bis(trifluoromethyl)phenyl phosphine oxide (T_g_ = 247 °C) [38].

### 3.3. Optical and Dielectric Properties

The optical properties, including the cutoff wavelength (λ), optical transmittance at the wavelength of 450 nm (T_450_), refractive indices, and CIE optical parameters of the PPI films were investigated and the results are summarized in Table 4. Figure 10a shows the UV-Vis spectra of the PPI films. Basically, the PPI films exhibited good optical transparency in the visible light region (400~760 nm) with the λ and T_450_ values of 349 nm and 82.8% for PPI-1 film and 376 nm and 78.6% for PPI-2 film, respectively. The PPI-2 film based on the rigid 6FCDA dianhydride showed a slightly lower optical transparency than that of its PPI-1 counterpart, which was mainly due to the condense packing of the molecular chains in the PPI-2 system. This structural feature was further reflected by the refractive index (n) results of the polymers. According to the Lorentz–Lorenz equation, the n value of one polymer is usually related to the molar refraction and molar volumes of the groups in the polymer [39]. Lower molar volumes or higher molecular chain packing density will usually result in higher n values for the polymer. As shown in Table 4, PPI-2 exhibited an average n (n_av_) value of 1.6280, which was higher than that of the PPI-1 (n_av_ = 1.6152). Meanwhile, the relatively higher packing density of the molecular chains increased the intermolecular interactions and the degree of conjugation. Thus, the PPI-2 film exhibited higher yellow index (b* = 9.04), lower lightness (L* = 94.87), and higher haze (1.16%) in the CIE optical measurements. Although the optical properties of the PPI-2 film were inferior to those of its PPI-1 counterpart, the optical values of the PPI-2 film could still meet the requirements for space applications.

The dielectric properties of the PPI films, including dielectric constant (D_k_) and dielectric dissipation factor (D_f_), were measured at a frequency of 10 GHz and the results are shown in Table 4. Incorporation of the hexafluoroisopropylidene groups with low molar polarization and the bulky DPPO substituents with large molar volume efficiently endowed the PPI films with moderate to low D_k_ and D_f_ values according to the Clausius–Mossotti equation [40]. As expected, the PPI-2 film showed D_k_ and D_f_ values of 3.44 and 0.0304, which were all higher than those of the PPI-1 film (D_k_ = 3.26; D_f_ = 0.0233). This was also due to the relatively lower molar volume of the molecular chains in PPI-2 caused by the planar and rigid xanthene units in 6FCDA moiety. The low-D_k_ and low-D_f_ features of the PPI films make them good candidates as dielectric components for space applications.

In summary, the PPI-2 film showed comparable optical and dielectric properties with those of the fluoro- and phosphorus-containing PI films reported in the literature, such as the one derived from 6FDA and bis(3-aminophenyl) methyl phosphine oxide (T_450_ = 88.2%; D_k_ = 3.27 (100 kHz); D_f_ = 0.0128 (100 kHz)) [41].

### 3.4. AO Erosion Properties

The AO erosion properties of the PPI films were investigated in a ground simulation facility with the AO irradiation fluence of 5.0 × 10^20^ atoms/cm^2^. The Kapton^®^ type of PI film was used as the standard with an AO erosion yield (E_y_) of 3.0 × 10^−24^ cm^3^/atom. It has been well established in the literature that the phosphorous-containing PI (PPI) films show good self-healing features in AO environments [42,43,44,45]. This is based on the mechanism that when the surface of the PPI films was attacked by AO, inert phosphorus oxide or phosphate passivation layers could in situ form and prevented the underlayer from further erosion by the AO irradiation. In the current work, the AO erosion properties of the PPI films were evaluated. The AO-eroded PPI films were named as PPI-1-AO and PPI-AO-2, respectively. The mass loss of the PPI films after AO exposure was recorded and the results are shown in Table 5. According to the mass loss, the AO erosion yields (E_y_) of the PPI films were calculated according to Equation (2) and the results are also listed in Table 5. It could be clearly seen that both PPI films showed obviously enhanced AO resistance compared with that of the standard Kapton^®^ reference. For example, the PPI films showed an E_y_ value of 6.99 × 10^−25^ cm^3^/atom for PPI-1 and 7.23 × 10^−25^ cm^3^/atom for PPI-2, respectively, which were only 23.3~24.1% of the erosion yield of Kapton film (E_y_ = 3.0 × 10^−24^ cm^3^/atom). In addition, the PPI-2 film showed a slightly higher E_y_ value than that of PPI-1, which might be due to the relatively lower phosphorus (P) content of the repeating units. The P contents of PPI-1 and PPI-2 were 3.44% and 3.39%, respectively. Higher P content in the PPI-1 film resulted in better AO resistance of the polymer.

AO irradiation not only caused the mass loss of the PPI films, but also had a significant impact on the optical properties of the films. Figure 10b–e qualitatively compare the appearance of the PPI films before and after AO irradiation. Apparently, the pristine PPI films (left samples in Figure 10b,c) showed clear and transparent appearance, while the AO-eroded PPI films (samples in Figure 10d,e) exhibited a translucent appearance. The optical transmittance of the PPI films before and after AO irradiation was further quantitatively compared as shown in Figure 10e. The AO irradiation greatly deteriorated the optical transmittance of the PPI films. For example, the pristine PPI-2 film showed a T_450_ value of 78.6%, while the T_450_ value of the PPI-2-AO film decreased to 4.4%.

The main reason for the deterioration of the optical transmittance of the PPI films might be ascribed to the formation of rough passivation layers on the surface of PPI films after AO irradiation. This could be qualitatively proven by the SEM and AFM measurements. Figure 11 shows the SEM and energy dispersive spectroscopy (EDS) of the PPI-1 and PPI-2, respectively. For both films, after AO irradiation, the originally smooth surfaces of the pristine PPI films became rough, forming a carpet-like micromorphology. Moreover, as can be seen from the EDS spectra, the element intensities and distribution in the surfaces of the PPI films also changed significantly before and after AO irradiation.

The roughness of the film surfaces, including average roughness (R_a_) and root mean square roughness (R_q_) was quantitatively evaluated by the AFM measurements and the results are shown in Figure 12. According to the three-dimensional (3D) AFM images of 5.0 µm × 5.0 µm for the PPI films, the R_a_ and R_q_ values of the films apparently increased after AO irradiation. For example, for the PPI-2 film, the R_a_ and R_q_ values increased from 22.8 nm and 26.7 nm to 227.0 nm and 270.0 nm, respectively. That is to say, the AO irradiation increased the roughness of the PPI-2 film surface by about 10 times. The increase in the surface roughness had strong scattering effects to the incident lights, which then reduced the optical transmittance of the film.

The surface elemental compositions of the pristine and AO-eroded PPI films were detected by XPS measurements and the results are shown in Figure 13 and Figure 14, and Table 6. As compared in the figures, the atomic concentrations of carbon (C1s) greatly decreased after AO irradiation, whereas for oxygen (O1s) and phosphorus (P2s and P2p) these dramatically increased. For example, for PPI-2 film, the atomic concentrations of C1s, O1s and P2p changed from 76.37%, 13.03% and 0.91% in the pristine film to 40.59%, 42.18%, and 10.40% in the AO-irradiated film, respectively. Comparatively, the atomic concentrations for nitrogen (N1s) and fluorine (F1s) changed little, indicating the potentially slight interactions of these elements with AO species during the irradiation.

Figure 15 illustrates the changes of the binding energy of the P2p and O1s of the PPI films before and after AO exposure. As shown in Figure 15a, PPI-1 film exhibited 2.69 eV increase in the binding energy of P2p (132.16→134.85) and 0.93 eV increase in the binding energy of O1s (532.04→532.97) after the AO irradiation. Similarly, the PPI-2 system showed binding energy increases of 2.55 eV for P2p and 0.77 eV for O1s after AO erosion, respectively (Figure 15b). In addition, the atomic concentration ratio of oxygen to phosphorus was around 4:1 for the AO-eroded PPI films. Generally, the shifts in the binding energy and the broadening of the absorption peaks indicated the formation of higher oxidized phosphorus species, such as phosphate [46]. The inert phosphate passivation layer efficiently prohibited further erosion of the underlayers in the PPI films.

## 4. Conclusions

Fluoro-containing PPI films with enhanced thermal stability, high-temperature dimensional stability, and good AO resistance were developed in the current work. The PPI-2 film based on the tricyclic 6FCDA dianhydride and the phosphorus-containing BADPO diamine exhibited good combined properties, including a T_g_ value of 316.9 °C (DMA), a T_5%_ value of 501.3 °C, a CTE value of 41.7 × 10^−6^/K, a T_S_ value of 125.5 MPa, a T_M_ value of 4.05 GPa, a T_450_ value of 78.6%, and an E_y_ value of 7.23 × 10^−25^ cm^3^/atom. The main problems for the PPI films were the severe deterioration in the optical transmittances of the films after AO irradiation. This drawback might greatly limit the applications of the current films as optical components for LEO spacecraft. Incorporation of colloidal silica nanoparticles into the PPI films might be an effective method to dissolve the problem. Once the dense, continuous, and transparent silica passivation layer formed on the PPI films after AO irradiation, the optical transmittance of the PPI films might be improved. This idea will be investigated in our future work.

## Figures and Tables

**Figure 1 polymers-16-00343-f001:**
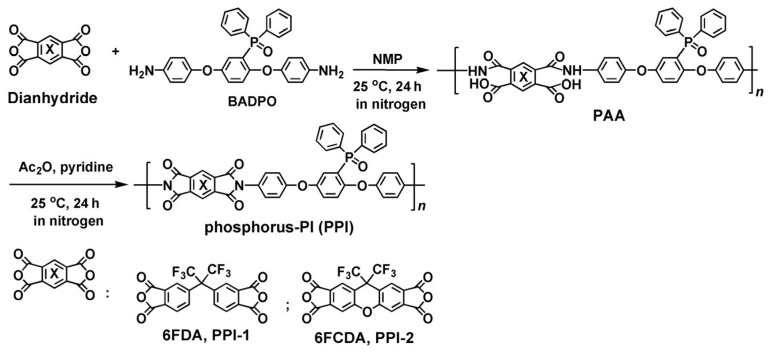
Preparation of PPI resins.

**Figure 2 polymers-16-00343-f002:**
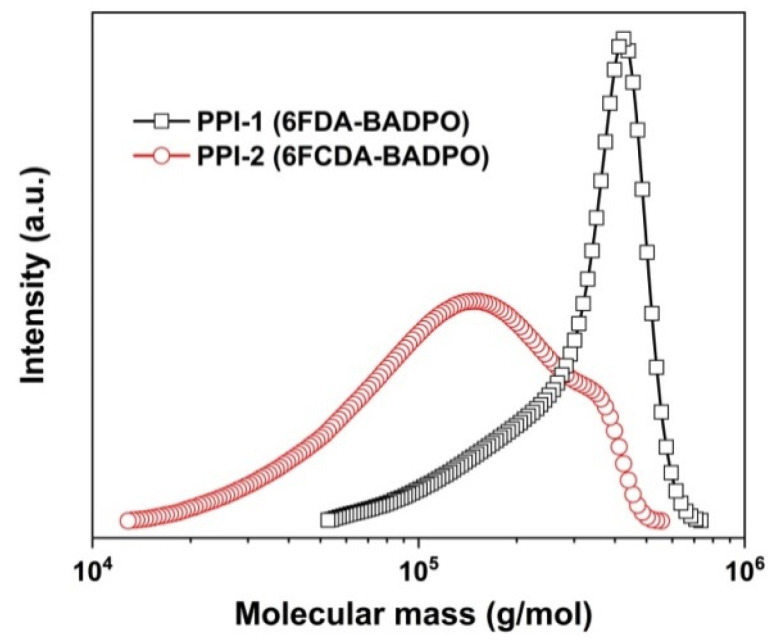
M_w_ values of PPI resins measured by GPC.

**Figure 3 polymers-16-00343-f003:**
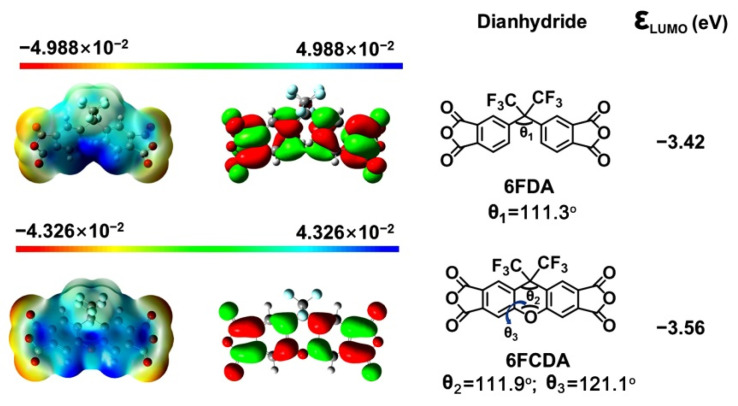
Molecular orbit energy levels (ε_LUMO_) of the dianhydrides.

**Figure 4 polymers-16-00343-f004:**
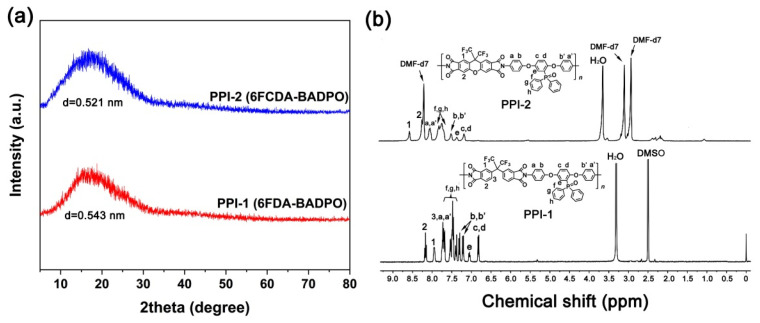
XRD spectra (**a**) and ^1^H-NMR spectra (**b**) of PPI resins.

**Figure 5 polymers-16-00343-f005:**
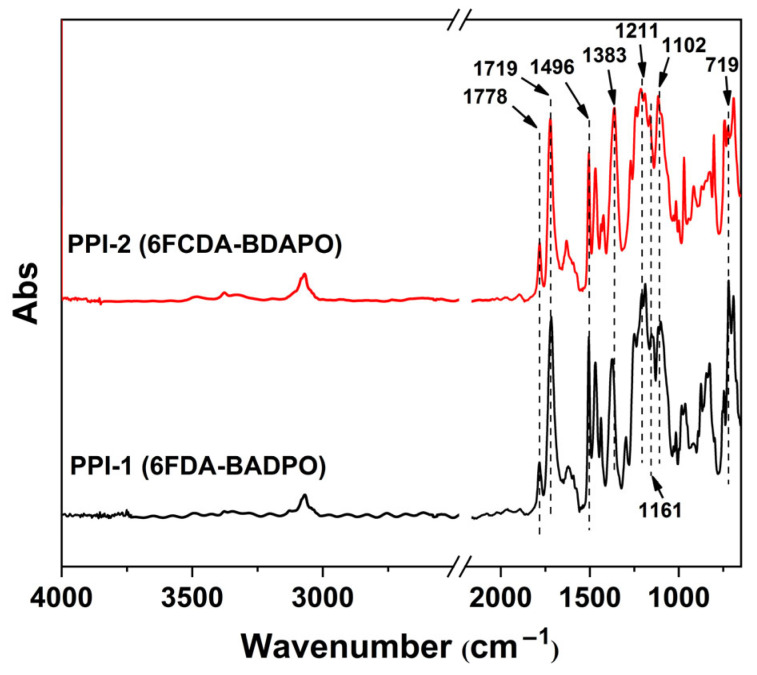
FTIR spectra of PPI films.

**Figure 6 polymers-16-00343-f006:**
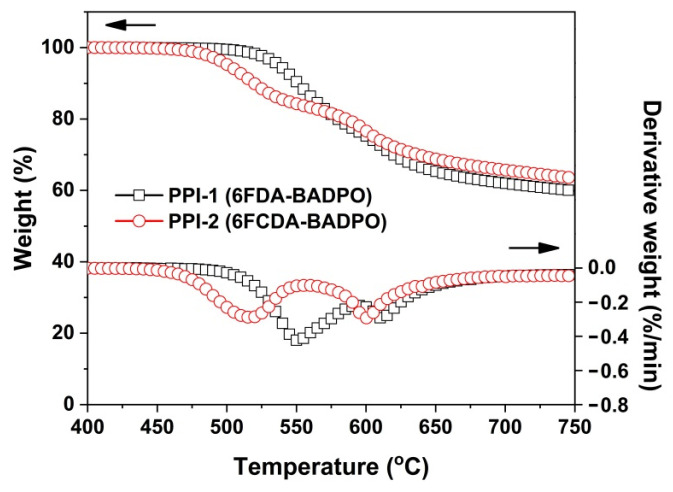
TGA and DTG curves of PPI films in nitrogen.

**Figure 7 polymers-16-00343-f007:**
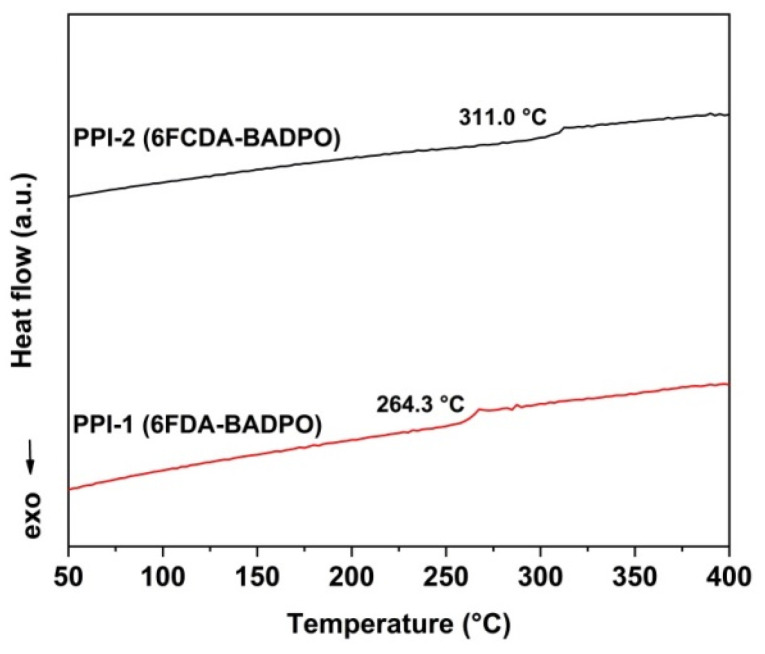
DSC curves of CPI films.

**Figure 8 polymers-16-00343-f008:**
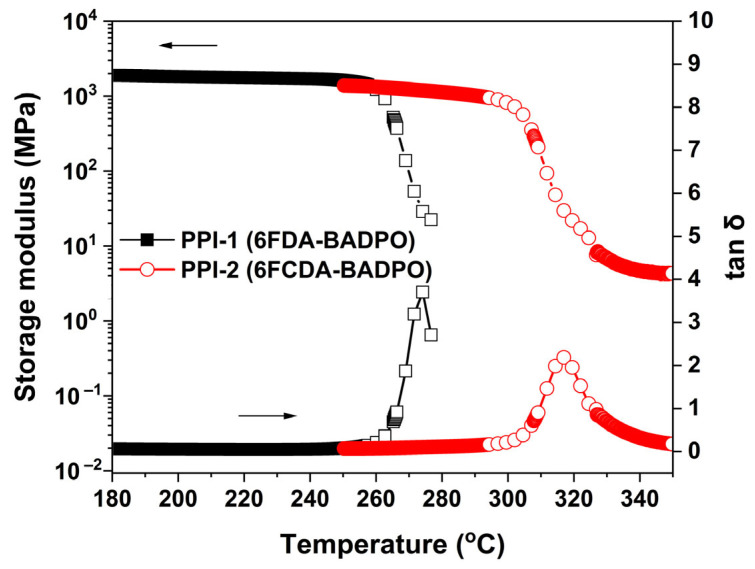
DMA curves of PPI films.

**Figure 9 polymers-16-00343-f009:**
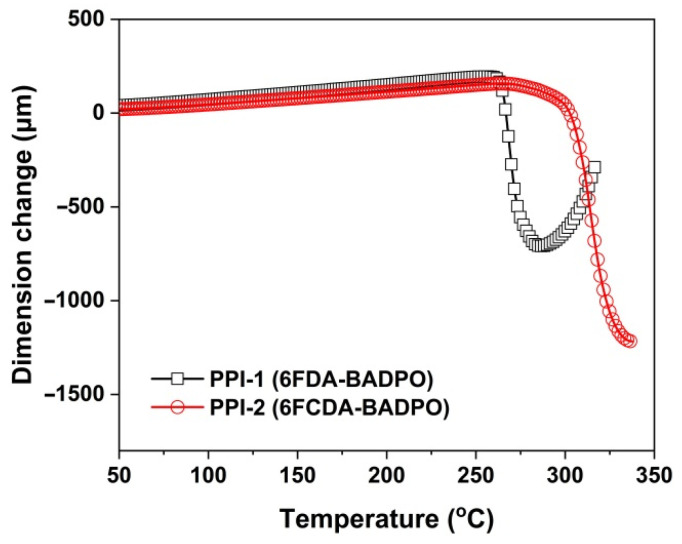
TMA curves of PPI films.

**Figure 10 polymers-16-00343-f010:**
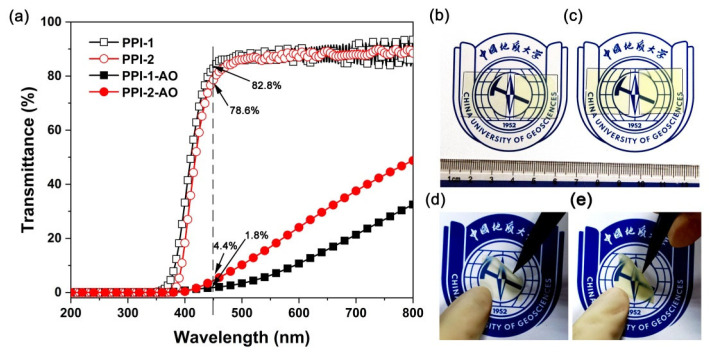
Optical transparency and appearance of the PPI films. (**a**) UV-Vis spectra; (**b**) PPI-1, left: pristine film; right: PPI-1-AO; (**c**) PPI-2, left: pristine film; right: PPI-2-AO; (**d**) PPI-1-AO; (**e**) PPI-2-AO.

**Figure 11 polymers-16-00343-f011:**
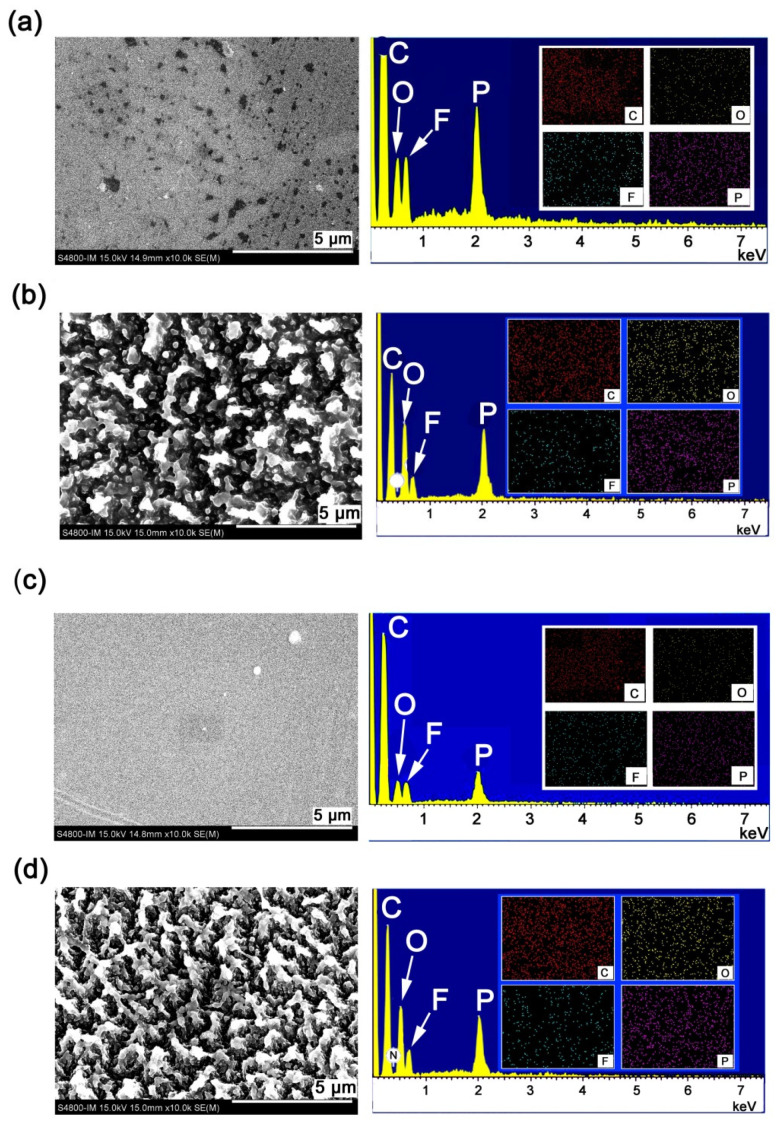
SEM and EDS images of PPI films. (**a**) PPI-1; (**b**) PPI-1-AO; (**c**) PPI-2; (**d**) PPI-2-AO.

**Figure 12 polymers-16-00343-f012:**
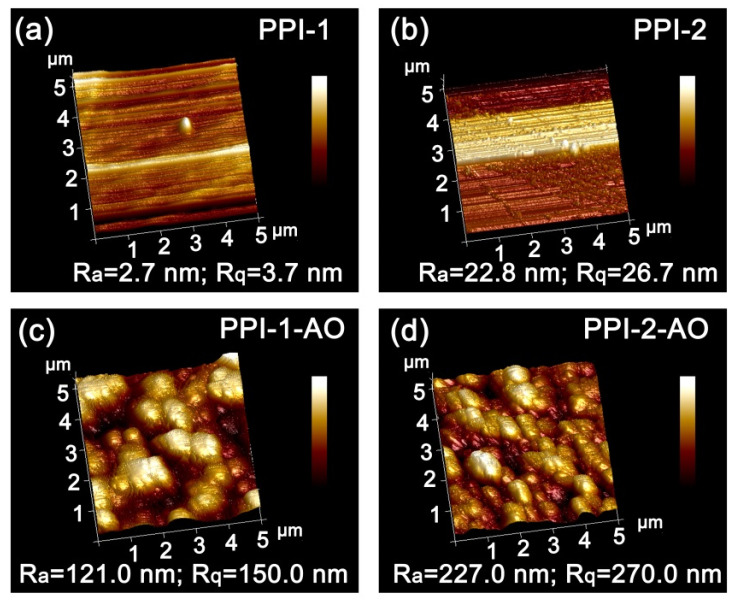
The 3D AFM images (5.0 µm × 5.0 µm) of PPI films after AO exposure (5.0 × 10^20^ atoms/cm^2^). (**a**) PPI-1, (**b**) PPI-2, (**c**) PPI-1-AO, (**d**) PPI-2-AO.

**Figure 13 polymers-16-00343-f013:**
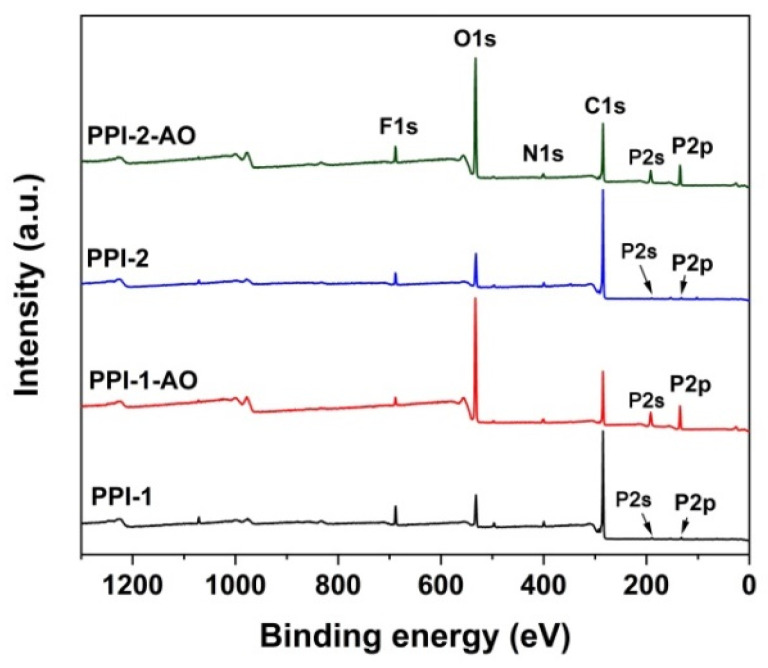
XPS spectra of PPI films before and after AO exposure.

**Figure 14 polymers-16-00343-f014:**
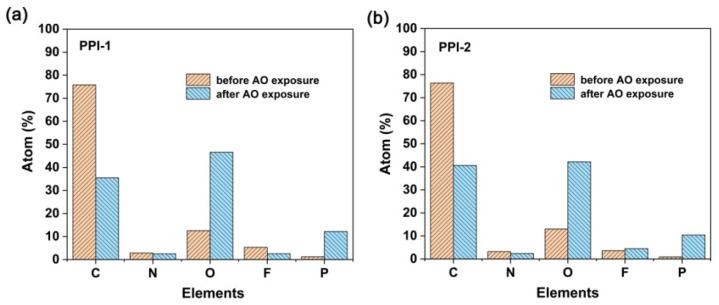
Atom percent of PPI films before and after AO exposure (fluence: 5.0 × 10^20^ atom/cm^2^). (**a**) PPI-1; (**b**) PPI-2.

**Figure 15 polymers-16-00343-f015:**
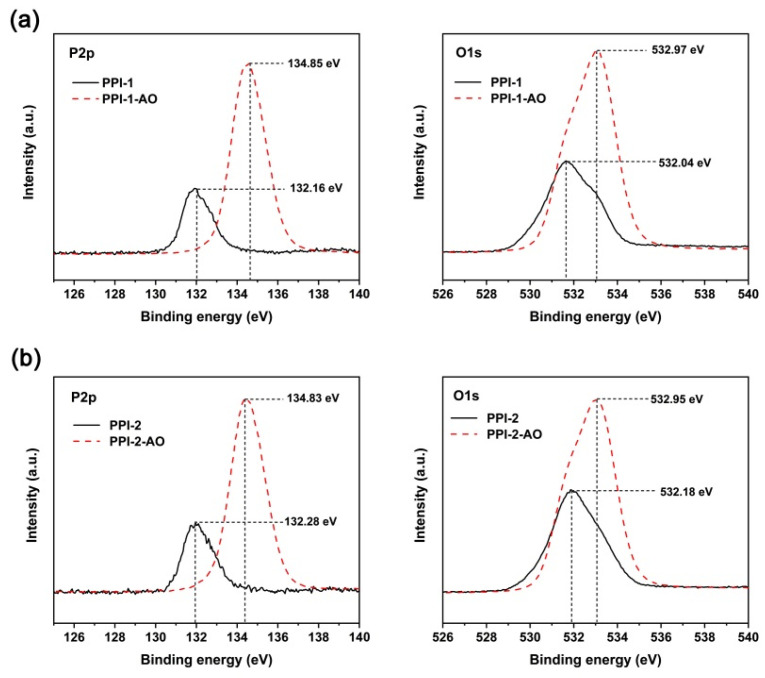
XPS spectra of P2p and O1s for PPI and AO-eroded PPI films. (**a**) PPI-1 and PPI-1-AO; (**b**) PPI-2 and PPI-2-AO.

**Table 1 polymers-16-00343-t001:** Formula for the PI and referenced PI resins synthesis.

PI	6FDA ^a^ (g, mol)	6FCDA ^a^ (g, mol)	BADPO ^a^ (g, mol)	NMP ^a^ (g)	Ac_2_O ^a^ (g)	Pyridine (g)
PPI-1	22.2120, 0.05	NA ^b^	24.6250, 0.05	140.5	30.6	19.8
PPI-2	NA	22.9110, 0.05	24.6250, 0.05	142.6	30.6	19.8

^a^ 6FDA: 4,4′-(hexafluoroisopropylidene)diphthalic anhydride; 6FCDA: 9,9-bis(trifluoromethyl)xanthene- 2,3,6,7-tetracarboxylic dianhydride; BADPO: 2,5-bis[(4-aminophenoxy)phenyl]diphenylphosphine oxide; NMP: N-methyl-2-pyrrolidone; Ac_2_O: acetic anhydride; ^b^ Not applicable.

**Table 2 polymers-16-00343-t002:** Inherent viscosities, molecular weights, and solubility of PPI resins.

PI	[η]_inh_ ^a^ (dL/g)	Molecular Weight ^b^			Solubility ^c^				
		M_n_ (×10^4^ g/mol)	M_w_ (×10^4^ g/mol)	PDI	NMP	DMAc	DMF	DMSO	CPA
PPI-1	1.38	26.74	33.74	1.26	++	++	++	++	++
PPI-2	1.21	9.61	15.27	1.59	++	++	++	+	−

^a^ Inherent viscosities measured with a 0.5 g/dL PI solution in NMP at 25 °C; ^b^ M_n_: number average molecular mass; M_w_: weight average molecular mass; PDI: polydispersity index, PDI = M_w_/M_n_; ^c^ ++: Soluble; +: partially soluble; −: insoluble. DMSO: dimethyl sulfoxide; CPA: cyclopentanone.

**Table 3 polymers-16-00343-t003:** Thermal and tensile properties of PPI films.

Samples	Thermal Properties ^a^							Tensile Properties ^b^		
	T_g, DSC_ (°C)	T_g, DMA_ (°C)	T_5%_ (°C)	T_max1_ (°C)	T_max2_ (°C)	R_w750_ (%)	CTE (×10^−6^/K)	T_S_ (MPa)	T_M_ (GPa)	ε (%)
PPI-1	264.3	274.2	537.6	550.5	611.1	59.9	49.2	115.1	3.54	6.4
PPI-2	311.0	316.9	501.3	516.0	600.1	63.4	41.7	125.5	4.05	9.9

^a^ T_g, DSC_: Glass transition temperatures according to the DSC measurements; T_g, DMA_: Glass transition temperatures according to the DMA measurements (peaks of tan δ plots); T_5%_: Temperatures at 5% weight loss; T_max1_, T_max2_: Temperatures corresponding to the first-stage and second-stage rapidest thermal decomposition rate, respectively; R_w750_: Residual weight ratio at 750 °C in nitrogen; CTE: linear coefficient of thermal expansion in the range of 50–250 °C. ^b^ T_S_: tensile strength; T_M_: tensile modulus; ε: elongation at break.

**Table 4 polymers-16-00343-t004:** Optical and dielectric properties of PPI films.

Samples	Optical Properties ^a^										Dielectric Properties ^b^	
	λ (nm)	T_450_ (%)	n_TE_	n_TM_	n_av_	Δn	L*	a*	b*	Haze (%)	D_k_	D_f_
PPI-1	349	82.8	1.6194	1.6068	1.6152	0.0126	95.39	−2.42	5.77	0.74	3.26	0.0233
PPI-2	376	78.6	1.6384	1.6069	1.6280	0.0315	94.87	−3.39	9.04	1.16	3.44	0.0304

^a^ λ: Cutoff wavelength; T_450_: Transmittance at the wavelength of 450 nm at a thickness of 25 µm; n_TE_, n_TM_: in-plane and out-of-plane refractive indices of the PPI films measured at 632.8 nm, respectively; n_av_: average refractive indices; Δn: birefringence, Δn = n_TE_ − n_TM_; L*, a*, b*, see Measurements; ^b^ D_k_, D_f_: dielectric constant and dielectric dissipation factor measured at the frequency of 10 GHz.

**Table 5 polymers-16-00343-t005:** Weight loss and erosion yields for PPI films.

Samples	W_1_ ^a^ (mg)	W_2_ ^a^ (mg)	ΔW ^a^ (mg)	E_y_ ^a^ (10^−25^ cm^3^/atom)
PPI-1	12.68	10.95	1.74	6.99
PPI-2	16.96	15.31	1.65	7.23
Kapton^®^	−	−	−	30.0

^a^ W_1_: Weight of the sample before irradiation; W_2_: Weight of the sample after irradiation; ΔW: Weight loss of the sample after AO irradiation, ΔW = W_1_ − W_2_; E_y_: Erosion yield.

**Table 6 polymers-16-00343-t006:** XPS results for the unexposed and exposed PI films.

Samples	Relative Atomic Concentration (%)									
	Unexposed Samples					AO Exposed Samples				
	C1s	N1s	O1s	F1s	P2p	C1s	N1s	O1s	F1s	P2p
PPI-1	75.83	2.87	12.55	5.35	1.22	35.46	2.50	46.57	2.55	12.17
PPI-2	76.37	3.20	13.03	3.65	0.91	40.59	2.37	42.18	4.46	10.40

## Data Availability

Data is contained within the article.

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
