# Peer review of "Preparation and Properties of Atomic-Oxygen Resistant Polyimide Films Based on Multi-Ring Fluoro-Containing Dianhydride and Phosphorus-Containing Diamine"

_polymers, 2024, doi:10.3390/polym16030343_

Round 1

Reviewer 1 Report

Comments and Suggestions for Authors

The manuscript is well-written, the results are in agreement with the proposed polymer structures. However there are some aspects that need clarification prior to publication, according to the comments below:

 INTRODUCTION

-        -   Row 81-91: in Introduction there is a paragraph describing the benefits given by utilization of 6FCDA dianhydride in polyimide structure, but no associated diamines are mentioned. These aspects must be presented in detail in the paper to emphasize the difference between the 6FCDA –based PI structures from literature and those prepared by your group.

-       -    in Introduction a short paragraph mentioning the originality of your work must be inserted. Also, the clear motivation of the study should be explained, including the choice of combining phosphorous and fluorine groups for your polyimide products.

-         -  Moreover in this introductive part of the paper, authors mention a general use of polyimides “different space applications” for “low earth orbit (LEO)” or “geosynchronous earth orbit (GEO)”, but they fail to specify the concrete application of the polyimides, and how the pursued properties in their paper (aside from thermal and mechanical properties) are useful for meeting the applicative criteria?

EXPERIMENTAL

-         - In experimental in “2.2. Characterization methods” authors obtained XRD data on polymer powder or on polymer film? Please clarify in the paper since the results are affected by this.

-       -   The reported data on the erosion yield are not so accurate since authors presume that their samples have similar density with Kapton. Why did you not determine the density of your polyimide samples?

-      -    Please provide all details regarding the AFM cantilever used for scans.

-      -    Why did authors prefer chemical imidization over thermal imidization route in polyimide preparation?

RESULTS AND DISCUSSION

-          - The FTIR data of polyimides should show the absence of peaks at about 3200 cm−1 and 1640 cm−1, corresponding to -NH and -CO stretching in the amide groups from polyamic acid (PAA) to indicate the completion of the imidization procedure (proper transformation of the PAA into the polyimide structure). How do authors comment the peaks at about 1600cm-1 and 3100 cm-1 from Fig. 6?

-       -   Row 340: table 3 caption needs completion given the displayed data.

-         - Arrows indicating the y-axis in Fig 9 must be inserted in the figure.

-        -  Row 398: table 4 caption needs completion given the displayed data.

-     -     Deeper explanation on the role of parameters from Table 4 (optical and dielectric properties) in the pursued space application must be introduced in the paper.

-     -     All thermal, mechanical, optical, dielectric data are well commented, but the paper needs to compare the results with those reported for other polyimides containing fluorine and/or phosphorous from literature to acquire for the reader a proper understanding. Proper comments must be included in the paper for each kind of property.

-       -   Some Figures should be merged since there are too many in the paper (example: 11 + 12; 13 + 14)

-     -     EDS and AFM images are too small and hence unclear. Please revise this!

-      -    How the groove-like morphology in Fig 15a &b can be explained? This is not a typical pristine polyimide surface morphology.

- 

Comments on the Quality of English Language

         There are few typos and English language errors that need correction.

Author Response

Response to reviewer 1:

  1. Question: Row 81-91: in Introduction there is a paragraph describing the benefits given by utilization of 6FCDA dianhydride in polyimide structure, but no associated diamines are mentioned. These aspects must be presented in detail in the paper to emphasize the difference between the 6FCDA –based PI structures from literature and those prepared by your group.

Answer: Thanks a lot. As suggestion, the paragraph was rewritten in our revised manuscript as follows.

…Various common aromatic diamines, including 4,4ʹ-oxydianiline (ODA), p-phenylenediamine (PPD), diaminodurene (DAD), 3,3ʹ,5,5ʹ-tetramethylbenzidine (TMB),  2,2ʹ-dimethylbenzidine (DMB),  2,2ʹ-bis(trifluoromethyl)benzidine (TFMB), and bis(4-aminophenyl)hexafluoropropane (6FDAm) were used for the development of 6FCDA-PIs. Nevertheless, to the best of our knowledge, until today, the application of 6FCDA dianhydride in FPIs has not been much more extensively studied, and the structure-property relationship of the 6FCDA-based FPIs with functional diamines remains to be revealed more deeply and systematically. ”.

  1. Question: in Introduction a short paragraph mentioning the originality of your work must be inserted. Also, the clear motivation of the study should be explained, including the choice of combining phosphorous and fluorine groups for your polyimide products.

Answer: Thanks a lot. As suggestion, the sentences were added in our revised manuscript as follows.

The purpose of the current work is to develop functional FPI films with potential applications in LEO space environments. The phosphorus-containing diamine might endow the FPI films with enhanced atomic-oxygen resistance, while the 6FCDA dianhydride might improve the glass transition temperatures (Tg) and decrease the linear coefficients of thermal expansion (CTE) of the FPI films.”.

  1. Question:Moreover in this introductive part of the paper, authors mention a general use of polyimides “different space applications” for “low earth orbit (LEO)” or “geosynchronous earth orbit (GEO)”, but they fail to specify the concrete application of the polyimides, and how the pursued properties in their paper (aside from thermal and mechanical properties) are useful for meeting the applicative criteria?

Answer: Thanks a lot. As suggestion, the sentences were added in our revised manuscript as follows.

Thus, functional PI films have been widely studied and developed so as to meet the property requirements for LEO, GEO or deep-space applications. For the LEO applications, such as the multi-layer thermally insulating (MLI) blankets for satellites, thermal shielding layers for the space telescopes, substrates for solar cells, and so on, the PI films should have excellent AO resistance beside the excellent thermal and mechanical properties. While for the GEO or deep-space applications, such as the membranes for solar sails, the optical components for satellites, and so on, the PI films should have excellent optical transparency, high thermal and dimensional stability and good long-term reliabilities [12]. ”.

  1. Question:In experimental in “2.2. Characterization methods” authors obtained XRD data on polymer powder or on polymer film? Please clarify in the paper since the results are affected by this.

Answer: The XRD data were obtained with polymer powder and the explanation was added in section 2.2 in our revised manuscript.

  1. Question:The reported data on the erosion yield are not so accurate since authors presume that their samples have similar density with Kapton. Why did you not determine the density of your polyimide samples?

Answer: Thanks a lot. The calculation method for the erosion yields of the PI films in the current work was commonly used in the literature, in which the density of the solid PI films (except the foamed or gelled ones) was assumed to be the similar one with the referenced Kapton film (repeating unit: C22H10O5N2, volume density: 1.42 g/cm3, ref: Acta Astronautica 207 (2023) 118–128) (Applied Surface Science 631 (2023) 157562; ACS Appl. Mater. Interfaces, 2012, 4, 492-502, et al.). Thus, we used the same method for the calculation of the erosion yields of the PI samples.

  1. Question:Please provide all details regarding the AFM cantilever used for scans.

Answer: The details regarding the AFM cantilever used for scans were added in section 2.2 in our revised manuscript as follows.

The Bruker Scanasyst-Air model of cantilever (Santa Babara, California, USA) with the resonance frequency (f0) of 70 kHz, spring constant (k) of 0.4 N/m and the size of 115 μm×× 25 μm × 650 nm) was used.”.

  1. Question:Why did authors prefer chemical imidization over thermal imidization route in polyimide preparation?

Answer: Thanks a lot. As suggestion, the “molar refractive index (P)” was modified as “molar refraction (P)” in our revised manuscript.

  1. Question:Why did authors prefer chemical imidization over thermal imidization route in polyimide preparation?

Answer: The thermal imidization of the poly(amic acid) (PAA) precursors has to be performed at elevated temperatures higher than 300 oC. Thus, the chemical imidization route was preferred in our work because the procedure could provide relatively lower curing temperatures for the fabrication of the PI films. This is beneficial for prohibiting the potential oxidation effects and deterioration of the optical transparency and yellow indices for the afforded PI films.

  1. Question:The FTIR data of polyimides should show the absence of peaks at about 3200 cm−1 and 1640 cm−1, corresponding to -NH and -CO stretching in the amide groups from polyamic acid (PAA) to indicate the completion of the imidization procedure (proper transformation of the PAA into the polyimide structure). How do authors comment the peaks at about 1600 cm-1 and 3100 cm-1 from Fig. 6?

Answer: We treated the PI films at 300 oC and then re-measured the FTIR spectra of the PI films. The initial absorptions around 3400~3500 cm-1, which might be due to the water uptakes disappeared. However, the peaks at 3100 cm-1 still existed, which could be ascribed to the unsaturated C-H absorptions in the benzene rings. The peaks at 1600 cm-1 could be contributed to C=C absorptions in the benzene rings. Similar results have also been reported in the literature, such as ref. 38 in our revised manuscript.

The new FTIR spectrum was added in our revised manuscript.

  1. Question:Row 340: table 3 caption needs completion given the displayed data.

Answer: Thanks a lot. The caption of Table 3 was modified as follows.

“Table 3. Thermal and tensile properties of PPI films.”.

  1. Question:Arrows indicating the y-axis in Fig 9 must be inserted in the figure.

Answer: Figure 9 was updated in our revised manuscript.

  1. Question:Row 398: table 4 caption needs completion given the displayed data.

Answer: The caption of Table 4 was modified as follows.

“Table 4. Optical and dielectric properties of PPI films.”.

  1. Question:Deeper explanation on the role of parameters from Table 4 (optical and dielectric properties) in the pursued space application must be introduced in the paper.

Answer: Thanks a lot. As suggestion, the explanation on the relationship between the optical and dielectric properties with the space applications was added in our revised manuscript as follows.

…Although the optical properties of the PPI-2 film were inferior to those of the PPI-1 counterpart, the optical values of the PPI-2 film could still meet the requirements for space applications.”.

…The low-Dk and low-Dfeatures of the PPI films make them good candidates as dielectric components for the space applications. ”.

  1. Question: All thermal, mechanical, optical, dielectric data are well commented, but the paper needs to compare the results with those reported for other polyimides containing fluorine and/or phosphorous from literature to acquire for the reader a proper understanding. Proper comments must be included in the paper for each kind of property.

Answer: As suggestion, the thermal and mechanical properties of the PI films were compared with the similar systems reported in the literature in our revised manuscript as follows.

In summary, the currently developed PPI-2 film showed higher Tg, higher TS and higher TM values than those of the fluoro- and phosphorus-containing PI films reported in the literature, such as the one derived from 6FDA and 2,5-bis[(4-amino-3- trifluoromethylphenoxy)phenyl]diphenylphosphine oxide (Tg=271 oC; TS=124.9 MPa; TM=1.9 GPa) [37] or the one from 6FDA and bis(3-aminophenyl)-3,5- bis(trifluoromethyl)phenyl phosphine oxide (Tg=247 oC) [38]. ”.

The optical and dielectric properties of the PI films were compared with the similar systems reported in the literature in our revised manuscript as follows.

In summary, the PPI-2 film showed comparable optical and dielectric properties with those of the fluoro- and phosphorus-containing PI films reported in the literature, such as the one derived from 6FDA and bis(3-aminophenyl) methyl phosphine oxide (T450=88.2%; Dk=3.27 (100 kHz); Df=0.0128 (100 kHz)) [41]. ”.

Corresponding references were also added.

  1. Question:Some Figures should be merged since there are too many in the paper (example: 11 + 12; 13 + 14)

Answer: Thanks a lot. As suggestion, figure 11 and figure 12 were merged. Figure 13 and figure 14 were merged in our revised manuscript.

  1. Question:EDS and AFM images are too small and hence unclear. Please revise this!

Answer: Thanks a lot. As suggestion, the EDS and AFM images were enlarged in our revised manuscript.

  1. Question:How the groove-like morphology in Fig 15a &b can be explained? This is not a typical pristine polyimide surface morphology.

Answer: Thanks a lot. The groove-like morphology in Fig 15a &b might be due to the film-preparing process in the work. A helical bar was used for the film coating. This structure might form grooves during the thermal curing of the wet films.

  1. Question:There are few typos and English language errors that need correction.

Answer: The mistakes were carefully modified in our revised manuscript.

Reviewer 2 Report

Comments and Suggestions for Authors

The manuscript titled “Preparation and Properties of Atomic-oxygen Resistant Polyimide Films with Improved Glass Transition Temperatures based on Multiring Fluorocontaining Dianhydride and Phosphorus-containing Diamine” describes preparation of a new PI film based on 6FCDA and a phosphorus-containing diamine, 2,5-bis[(4-aminophenoxy)phenyl]diphenylphosphine oxide (BADPO), which was named as PPI-2. The topic is relevant and the manuscript deserves consideration. However, there are some point that can be fixed and improve the quality of the manuscript. The main comments and recommendations are listed below.

Title should be revised to make it shorter and understandable.

Details for chemicals, equipment and software should be unified: model/mark (Manufacturer, City, Country).

Table 1. All abbreviations should be defined under the table. Tables and figures should be readable separately from the text.

The same for other tables.

Why did the authors determine only LUMO of dianhydrides? What about HOMO? Did the authors simulate molecular complexes with determination of Chemical energy and chemical rigidity? This is needed for full understanding.

Since the figures mainly are not large, the authors can combine them as, for example, characterization of PPI resins and mark hem as a,b,c,d…This will make easier for readers to compare all parameters of characterization together in the same place

Figures 13,14,15 should be enlarged. SEM is bad visually presented, EDS maps are not readable, AFM axes are no readable. For example, the authors can check https://doi.org/10.1038/s41598-022-16878-w.

Figure 18. Where is “before” and where is “after”? The authors can use a1,a2, b1,b2 or just write above or below “before” and “after”

Discussion part should be improved with references to recent relevant works.

The manuscript should be carefully checked for typos and grammatical errors, there are many cases.

Comments on the Quality of English Language

The manuscript should be carefully checked for typos and grammatical errors, there are many cases.

Author Response

Response to reviewer 2:

  1. Question: Title should be revised to make it shorter and understandable.

Answer: Thanks a lot. The title was modified in our revised manuscript as follows.

Preparation and Properties of Atomic-oxygen Resistant Polyimide Films based on Multi-ring Fluoro-containing Dianhydride and Phosphorus-containing Diamine”.

  1. Question: Details for chemicals, equipment and software should be unified: model/mark (Manufacturer, City, Country).

Answer: The details for chemicals, equipment and software were unified in our revised manuscript.

  1. Question: Table 1. All abbreviations should be defined under the table. Tables and figures should be readable separately from the text. The same for other tables.

Answer: All abbreviations were added under the table in our revised manuscript.

  1. Question: Why did the authors determine only LUMO of dianhydrides? What about HOMO? Did the authors simulate molecular complexes with determination of Chemical energy and chemical rigidity? This is needed for full understanding.

Answer: Thanks a lot. The reaction mechanism for the formation of poly(amic acid) is the nucleophilic attack reaction of the amino groups in diamine to the anhydride carbonyl groups. Meanwhile, the charge transfer route in the PI is from the diamine units (HOMO orbits) to the dianhydride units (LUMO orbits. Thus, when we justify the polymerization reactivity of the monomers, the LUMO energy for dianhydride and the HOMO energy for diamine are more meaningful. Thus, only the LUMO energy of the dianhydride was determined in our work. Similar methodology was also used in the literature, such as ref. 32. The molecular complexes with determination of chemical energy and chemical rigidity were not simulated in the current work. Instead, the torsion angles of the dianhydrides were calculated. In our future work, the molecular complexes might be calculated and discussed.

  1. Question: Since the figures mainly are not large, the authors can combine them as, for example, characterization of PPI resins and mark hem as a,b,c,d…This will make easier for readers to compare all parameters of characterization together in the same place.

Answer: Thanks a lot. As suggestion, the XRD spectra (figure 4) and the NMR spectra (figure 5) were merged in our revised manuscript.

  1. Question: Figures 13,14,15 should be enlarged. SEM is bad visually presented, EDS maps are not readable, AFM axes are no readable. For example, the authors can check https://doi.org/10.1038/s41598-022-16878-w.

Answer: Thanks a lot. As suggestion, the figures were enlarged in our revised manuscript.

  1. Question: Figure 18. Where is “before” and where is “after”? The authors can use a1,a2, b1,b2 or just write above or below “before” and “after”.

Answer: As suggestion, the caption of Figure 18 was modifed in our revised manuscript.

  1. Question: Discussion part should be improved with references to recent relevant works.

Answer: Thanks a lot. The discussion part was modified in our revised manuscript.

  1. Question: The manuscript should be carefully checked for typos and grammatical errors, there are many cases.

Answer: The typos and grammatical errors were carefully revised in our revised manuscript.

Round 2

Reviewer 1 Report

Comments and Suggestions for Authors

The revised manuscript is acceptable for publication.

Reviewer 2 Report

Comments and Suggestions for Authors

The authors considered all comments and recommendation and decided them well or gave detailed explanation why their decisions are appropriate. The revised manuscript deserves further consideration by the Editor